# Pattern of COVID-19 in Sichuan province, China: A descriptive epidemiological analysis

Hongfei Song[1☯], Xiaoren Cao[2☯], Hua Ye[3☯], Li He[3☯], Guiyu Li[1], Tingjun Wan[1], Dong Wang[1], Yuqiao Liu[4], Zonghai Huang[3], Baixue Li[1], Li Wen[1], Yue Su[1], Cen Jiang[1]*, Quansheng Feng[1]*

1 School of Basic Medical Sciences, Chengdu University of Traditional Chinese Medicine, Chengdu, Sichuan, China, 2 Doctor of Department of Rehabilitation Medicine, Second people's Hospital of Ya'an City, Ya'an City, Sichuan, China, 3 School of Medical Information Engineering, Chengdu University of Traditional Chinese Medicine, Chengdu, Sichuan, China, 4 School of Clinical Medical, Chengdu University of Traditional Chinese Medicine, Chengdu, Sichuan, China

☯ These authors contributed equally to this work.
* fengqs118@163.com (QF); jiangcen517@163.com (CJ)

**Data Availability Statement:** All relevant data are within the manuscript and its Supporting Information files.

**Funding:** This work was supported by the National Major Project of Science and Technology (No.

## Abstract

This study described the epidemiology of 487 confirmed coronavirus disease 2019 (COVID-19) cases in Sichuan province of China, and aimed to provide epidemiological evidence to support public health decision making. Epidemiological information of 487 COVID-19 cases were collected from the official websites of 21 districts (including 18 cities, 3 autonomous prefecture) health commissions within Sichuan between 21st of January 2020 to 17th of April 2020. We focus on the single-day diagnosis, demographics (gender and age), regional distribution, incubation period and symptoms. The number of single-day confirmed COVID-19 cases reach a peak on January 29 (33 cases), and then decreased. Chengdu (121 cases), Dazhou (39 cases) Nanchong (37 cases) and Ganzi Tibetan Autonomous Prefecture (78 cases) contributed 275 cases (56.5% of the total cases) of Sichuan province. The median age of patients was 44.0 years old and 52.6% were male. The history of living in or visiting Hubei, close contact, imported and unknown were 170 cases (34.9%), 136 cases (27.9%), 21 cases (4.3%) and 160 cases (32.9%) respectively. The interval from the onset of initial symptoms to laboratory diagnosis was 4.0 days in local cases, while that of imported cases was 4.5 days. The most common symptoms of illness onset were fever (71.9%) and cough (35.9%). The growth rate of COVID-19 in Sichuan has significantly decreased. New infected cases have shifted from the living in or visiting Wuhan and close contact to imported. It is necessary to closely monitor the physical condition of imported cases.

## Introduction

Since December, 2019, Wuhan, a city of China, has reported an outbreak of atypical pneumonia caused by Severe Acute Respiratory Syndrome Coronavirus 2 (SARS-CoV-2) [1]. On 20th of January 2020, National Health Commission of the People's Republic of China issued a notice: cases of COVID-19 will be included in the Class B infectious diseases stipulated in the

2017ZX10205501), Industrial Cluster Collaborative Innovation Project of Chengdu Technology Bureau (2016-XT00-00033-GX), Study on the Biological Mechanism of "different treatment of the same Disease" in typical Syndromes of chronic Hepatitis B based on the differentiation of T Cell subsets mediated by H2b acetylation / HMGB1-PTEN Pathway (81803976) and Sichuan Science and Technology Innovation (Miaozi Project) cultivation Project in 2019049 (2019). The funders had no role in study design, data collection and analysis, decision to publish, or preparation of the manuscript. No authors received a salary from any of our funders.

**Competing interests:** The authors declare that they have no competing interests.

*Law of the People's Republic of China on the Prevention and Treatment of Infectious Diseases* and take measures for the prevention and control of Class A infectious diseases [2]. On 31st of January 2020, the World Health Organization (WHO) officially listed COVID-19 (Named by WHO since 11th of February 2020) as a public health emergency of international concern [3]. As of April 17, 2020, more than 2,034,802 confirmed cases were reported worldwide, and more than 135,163 infected patients died [4].

According to the National Health Commission of the People's Republic of China and the National Administration of Traditional Chinese Medicine *on Printing and Distributing a New Coronary Virus Pneumonia Diagnosis and Treatment Plan (Trial Version 6)* guidelines, the main clinical symptoms of COVID-19 are including fever, fatigue, dry cough, and a few patients also have a stuffy nose, runny nose, diarrhea, and other symptoms [5].

Currently, the pandemic of COVID-19 is moving rapidly around the world. Recently published research regarding the epidemiology of COVID-19 mainly concentrate on Wuhan city, but epidemiological investigations of other cities in China were rare. In this study, we explore epidemiological feature of Sichuan province to provide evidence for the formulation of public health strategies.

## Materials and methods

### Statistical analysis

We collected 487 confirmed COVID-19 cases on the official websites of the health committees of 21 districts in Sichuan province from 21st of January 2020 to 17th of April 2020. Available at: http://cdwjw.chengdu.gov.cn/; http://www.zg.gov.cn/web/swsjsw; http://wjw.panzhihua.gov.cn/; http://wjw.luzhou.gov.cn/; https://wjw.deyang.gov.cn/; http://wjw.my.gov.cn/; http://wsjsw.cngy.gov.cn/; http://swjw.suining.gov.cn/; http://wsj.neijiang.gov.cn/; http://swjj.leshan.gov.cn/swjj/index.shtml; http://wsjsw.nanchong.gov.cn/; http://ybwjw.yibin.gov.cn/; http://wjw.guang-an.gov.cn/; http://wjw.dazhou.gov.cn/; http://wsjkw.cnbz.gov.cn/index.html; http://wjw.yaan.gov.cn/; http://swjw.ms.gov.cn/; http://swjw.ziyang.gov.cn/; http://wjw.abazhou.gov.cn/; http://wjw.gzz.gov.cn/; http://www.lsz.gov.cn/ztzl/rdzt/yqfk/yqtb/.

Statistical analysis was performed by SPSS software (Version 21.0) and GraphPad Prism (Version 8.0.2). All patient data were fully anonymized before we accessed them.

## Results

### Demographic and clinical characteristics

This study comprised of 256 males (52.6%) and 231 females (47.4%), with a gender ratio of 1.1 to 1 (Table 1). The median age of patients was 44.0 age (ranged from 47 days to 88 years old; 95% CI: 41.2 to 44.1). The confirmed cases of aged 18–40 reached 191 cases (39.2%), and 41–65 reached 234 (48.0%). Aged 0–17 and over 66 year only had 62 cases (12.7%).

There were 170 cases with a history of living in or visiting Hubei, 136 cases of close contact history, 21 imported cases and 160 cases were unknown. The interval from the onset of initial symptoms to laboratory diagnosis was 4.0 days in local cases, while that of imported cases was 4.5 days. The most common symptoms of illness onset were fever (71.9%), cough (35.9%), and fatigue (9.8%). Gastrointestinal symptoms were uncommon (4.3%).

### Epidemiological situation

As of 17th of April 2020, there were a total of 561 laboratory confirmed cases in Sichuan province. We obtained data on the epidemiological characteristics of 487 patients (Fig 1). The

**Table 1. The demographic and clinical characteristics of Sichuan province.**

| Epidemiological characteristics | All patients (n = 487) |
|---|---|
| **Age** | |
| Median (IQR)—yr | 44.0 (30.0–53.0) |
| Distribution—no./total no. (%) | |
| 0 - 6yr | 9/487 (1.8) |
| 7 - 17yr | 13/487 (2.7) |
| 18 - 40yr | 191/487 (39.2) |
| 41 - 65yr | 234/487 (48.0) |
| ≥ 66yr | 40/487 (8.2) |
| **Gender** | |
| Male—no./total no. (%) | 256/487 (52.6) |
| **Exposure history** | |
| Live in or visit Hubei | 170/487 (34.9) |
| Close contact | 136/487 (27.9) |
| Imported | 21/487 (4.3) |
| Unknown | 160/487 (32.9) |
| **Symptoms—no.(%) (n = 256)** [§] | |
| Fever | 184/256 (71.9) |
| Chills | 17/256 (6.6) |
| Cough | 92/256 (35.9) |
| Sputum production | 20/256 (7.8) |
| Nasal congestion, runny or sneezing | 7/256 (2.7) |
| Sore throat | 15/256 (5.9) |
| Tightness in chest | 5/256 (2.0) |
| Shortness of breath or dyspnea | 13/256 (5.1) |
| Headache or dizziness | 15/256 (5.9) |
| Myalgia or arthral | 9/256 (3.5) |
| Fatigue | 25/256 (9.8) |
| Nausea or other stomach discomfort | 4/256 (1.6) |
| Diarrhea | 7/256 (2.7) |

Note: IQR, interquartile range.

[§] Data regarding symptoms were missing for 231 cases (47.4%).

numerator shows the cases included in our study cohort, and the denominator shows the number of laboratory confirmed cases in 21 districts of Sichuan province.

The first COVID-19 in Sichuan province was confirmed on 21st of January 2020 (in Chengdu). The amount of single-day newly diagnosed patients reached 33 on 29th of January and then showed a downward trend. After 29th of January, the rate of single-day newly patients decreased (Fig 2A). Since 28th of February, the number of newly confirmed patients slightly increased. The largest single-day increase of confirmed cases was in Chengdu, followed by Ganzi Tibetan Autonomous Prefecture, and the least was in Aba Tibetan and Qiang Autonomous Prefecture (Fig 2B).

Chengdu (121 cases), Dazhou (39 cases) Nanchong (37 cases) and Ganzi Tibetan Autonomous Prefecture (78 cases) contributed 275 cases (56.5% of the total cases) of Sichuan province. Excluding the unknown, the exposure history in Chengdu and Ganzi Tibetan

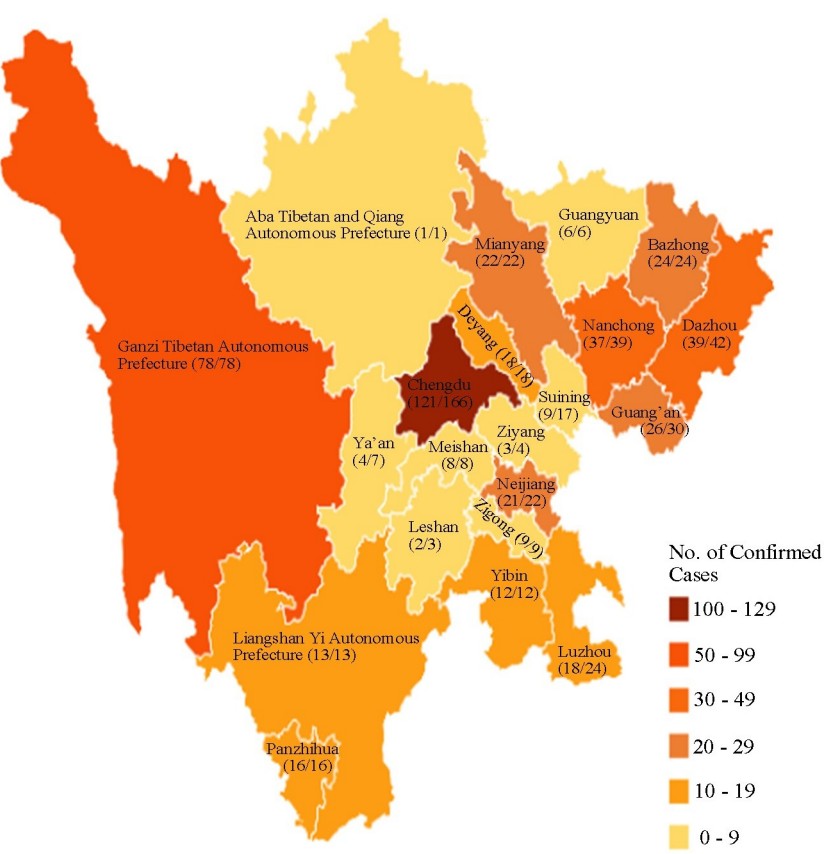

**Fig 1. Distribution of COVID-19 cases in Sichuan province.**

Autonomous Prefecture mainly concentrate on close contact history (57.4%) (Fig 2C). As a metropolis, Chengdu contributes to all imported cases (21 cases).

## Discussion

Our current study analysis the epidemiological pattern of SARS-CoV-2 infection in Sichuan province between 21st of January 2020 to 17th of April 2020. As of June 3rd, 2020, the confirmed COVID-19 cases of Sichuan province ranked 14th in China, with a cure rate of 96.7% and a case fatality rate of 0.5% [6].

A research covering 72,314 confirmed cases showed that the male-to-female ratio of confirmed cases was 0.99:1 in Wuhan, and 1.06:1 in China [7]. Combined with the results of the recent census in Sichuan Province (52.4% are male) [8], there was no distinct difference between men and women in this work, which was not in consistent with some current reports [9–11]. Therefore, it is necessary to conduct a larger, multicenter epidemiological survey to study whether there is gender susceptibility in COVID-19. In this research, most of the patients were young and middle-aged, and 34.9% of them had an exposure history in Wuhan. Specifically, in January 2020, most of the confirmed cases had an exposure history to Wuhan. This may be explained by population migration that Sichuan is a major labor export province in China. In February, the situation was a little different. Many confirmed cases were due to the close contact. It warned that the centralized isolation measures must be taken to hold back the recurrence of cluster epidemics. When it comes to March and April, most of these cases came from the United Kingdom and many were made up of students.

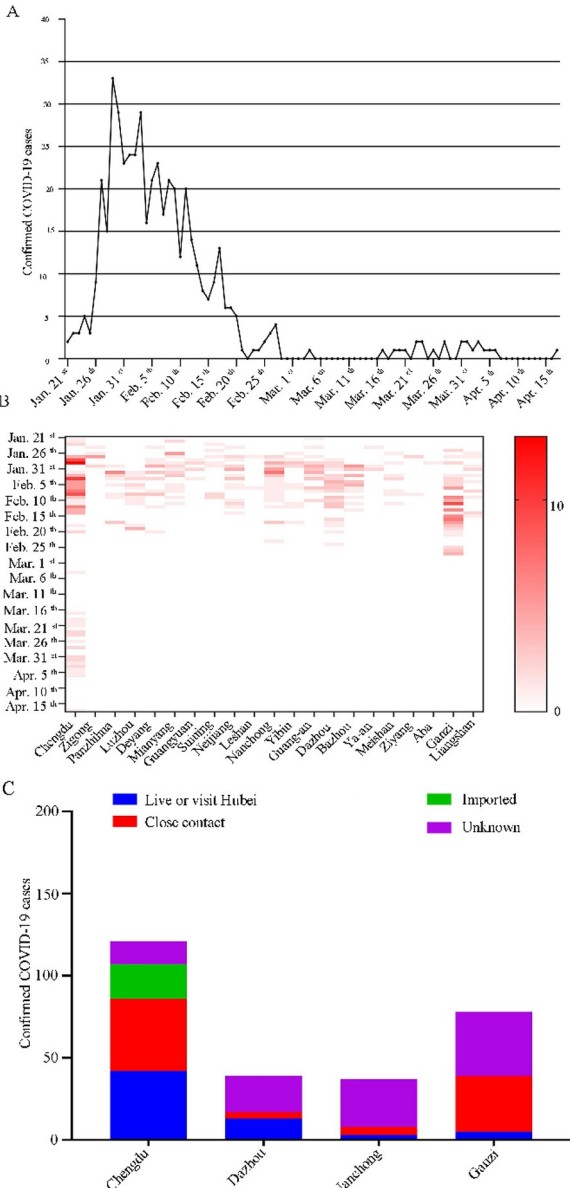

**Fig 2.** (A) Daily trend of 487 confirmed COVID-19 cases in Sichuan province from 21st of January 2020 to 17th of April 2020. (B) The heat-map shows the change of single-day confirmed cases in 21 districts of Sichuan province. (C) Exposure history of confirmed cases in Chengdu, Dazhou, Nanchong and Ganzi Tibetan Autonomous Prefecture.

The affected areas were unevenly distributed, mainly in the capital city of Chengdu, eastern Sichuan (Nanchong and Dazhou), which has a large population mobility. What is worth noting is that in this research many of the exposure history of confirmed cases in the Ganzi Tibetan Autonomous Prefecture are still unknown. It has been reported that agglomeration activities of family and religion have become the high-risk transmission factors of the epidemic in Ganzi Tibetan Autonomous Prefecture [12].

The interval from the onset of initial symptoms to laboratory diagnosis was 4.0 days in local cases, while that of imported cases was 4.5 days. Fever, cough and fatigue were the most common clinical symptoms, which was in consistent with the published reports [13, 14]. In this

report, there were 11 cases with gastrointestinal symptoms, which may be ignored. The existing documentation indicated that compared with patients with respiratory symptoms merely, patients with digestive tract symptoms usually had a longer course of disease between symptom onset and virus clearance, which may bring serious concequences to their contacters and themselves [15, 16]. Therefore, clinicians must keep in mind that when patients (such as those exposed to COVID-19) develop symptoms like fever and gastrointestinal, they should be caution to deal.

This study illustrates the epidemic of COVID-19 in Sichuan Province. Nevertheless, there are some limitation in this research. First, from 21$^{st}$ of January 2020 to 17$^{th}$ of April 2020, the official websites of Sichuan province recorded 561 cases, but age and gender were not reported in some cases. Therefore, our study included 487 coronavirus cases. Second, the quality of our data cannot support the estimation of the incubation period. In the next study, we plan to get more incubation period data and use reasonable model to evaluate the incubation period, so as to provide more information for the scientific prevention of COVID-19 in Sichuan Province.

## Supporting information

**S1 File.**
(PDF)

## Author Contributions

**Conceptualization:** Hongfei Song, Li He.

**Data curation:** Hongfei Song, Tingjun Wan.

**Formal analysis:** Xiaoren Cao, Hua Ye.

**Investigation:** Yuqiao Liu.

**Methodology:** Zonghai Huang.

**Project administration:** Baixue Li, Li Wen.

**Supervision:** Guiyu Li, Cen Jiang.

**Validation:** Dong Wang, Yue Su.

**Visualization:** Hongfei Song, Xiaoren Cao.

**Writing – original draft:** Hongfei Song, Li Wen.

**Writing – review & editing:** Quansheng Feng.

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
