## [Decision Letter · Decision Letter 0]

23 Apr 2020

PONE-D-20-09556

Pattern of COVID-19 in Sichuan Province, China: a descriptive epidemiological analysis

PLOS ONE

Dear Mr Feng,

Thank you very much for submitting your manuscript "Pattern of COVID-19 in Sichuan Province, China: a descriptive epidemiological analysis" (#PONE-D-20-09556) for review by PLOS ONE. As with all papers submitted to the journal, your manuscript was fully evaluated by academic editor (myself) and by independent peer reviewers. The reviewers appreciated the attention to an important health topic, but they raised substantial concerns about the paper that must be addressed before this manuscript can be accurately assessed for meeting the PLOS ONE criteria. Therefore, if you feel these issues can be adequately addressed, we invite you to submit a revised version of the manuscript that addresses the points raised during the review process. We can’t, of course, promise publication at that time.

We would appreciate receiving your revised manuscript by Jun 07 2020 11:59PM. To enhance the reproducibility of your results, we recommend that if applicable you deposit your laboratory protocols in protocols.io, where a protocol can be assigned its own identifier (DOI) such that it can be cited independently in the future. For instructions see: http://journals.plos.org/plosone/s/submission-guidelines#loc-laboratory-protocols

We look forward to receiving your revised manuscript.

Kind regards,

Abdallah M. Samy, PhD

Academic Editor

PLOS ONE

2. In the Methods, please provide a link to the official websites of the Health Commission in Sichuan, and/or describe how others may gain access to the data analysed in this study.

In addition, in the Methods, please state whether all patient data were fully anonymized before you accessed them.

3.Thank you for stating the following financial disclosure:

**Reviewers' comments:**

Reviewer's Responses to Questions

**Comments to the Author**

1. Is the manuscript technically sound, and do the data support the conclusions?

Reviewer #1: Yes

2. Has the statistical analysis been performed appropriately and rigorously? 

Reviewer #1: I Don't Know

3. Have the authors made all data underlying the findings in their manuscript fully available?

Reviewer #1: No

4. Is the manuscript presented in an intelligible fashion and written in standard English?

Reviewer #1: Yes

5. Review Comments to the Author

Reviewer #1: The study gives a descriptive analysis of 406 patients with SARS-CoV-2 infection in Sichuan province, China. I think the study can be accepted for publication because it gives some additional data on the COVID-19 in China. However, I am surprised to see very long incubation periods (median 12.5 days) among 40 patients who recalled their dates of exposure. For me, it is difficult to believe, because there are quite consistent estimates existent in the literature of about 5 days as the median. I suspect there is something wrong with the recalled dates: maybe the dates were with some uncertainty and the authors didn't account for this in their analysis. Maybe the selection of those 40 patients had some bias and maybe some of them were exposed later in time (~multiple exposure). I would still not follow the main recommendation of the authors to extend the quarantine length of 14 days to 28 days. Because these issues are not well-addressed in the manuscript, and there are no details on selected/excluded patients, I don't think that the present study will be accepted in the research community.

L87: What exactly are those 21 official websites? Are those from subprovinces/cities of Sichuan province? It would good to specify it.

L157: "This study tried to explore the incubation period of coronavirus cases." the verb "tried" is not good for using in scientific literature. I advice just to write something like "This study explored..."

6. PLOS authors have the option to publish the peer review history of their article (what does this mean?). If published, this will include your full peer review and any attached files.

Reviewer #1: No

---

## [Author Response · Author response to Decision Letter 0]

26 Jun 2020

Respond to editor,

Thank you for your kindness. We have performed the suggested analyses and revised the manuscript according to your editorial comments. We hope that our revised manuscript is in an appropriate format for publication in PLOS ONE.

Respond to editor,

I would like to thank the reviewers for their careful work. After re-evaluating all cases and establishing exclusion criteria, we found that the median incubation period was 7 days, which was within the range of current reports. Thanks again to the reviewers for their careful work.

---

## [Decision Letter · Decision Letter 1]

13 Aug 2020

PONE-D-20-09556R1

Pattern of COVID-19 in Sichuan Province, China: a descriptive epidemiological analysis

PLOS ONE

Dear Dr. Feng,

Thank you very much for submitting your manuscript "Pattern of COVID-19 in Sichuan Province, China: a descriptive epidemiological analysis" (#PONE-D-20-09556R1) for review by PLOS ONE. As with all papers submitted to the journal, your manuscript was fully evaluated by academic editor (myself) and by independent peer reviewers. The reviewers appreciated the attention to an important health topic, but they raised substantial concerns about the paper that must be addressed before this manuscript can be accurately assessed for meeting the PLOS ONE criteria. Therefore, if you feel these issues can be adequately addressed, we invite you to submit a revised version of the manuscript that addresses the points raised during the review process. We can’t, of course, promise publication at that time.

We look forward to receiving your revised manuscript.

Kind regards,

Abdallah M. Samy, PhD

Academic Editor

PLOS ONE

**Reviewers' comments:**

Reviewer's Responses to Questions

**Comments to the Author**

1. If the authors have adequately addressed your comments raised in a previous round of review and you feel that this manuscript is now acceptable for publication, you may indicate that here to bypass the “Comments to the Author” section, enter your conflict of interest statement in the “Confidential to Editor” section, and submit your "Accept" recommendation.

Reviewer #1: (No Response)

2. Is the manuscript technically sound, and do the data support the conclusions?

Reviewer #1: Partly

3. Has the statistical analysis been performed appropriately and rigorously? 

Reviewer #1: No

4. Have the authors made all data underlying the findings in their manuscript fully available?

Reviewer #1: Yes

5. Is the manuscript presented in an intelligible fashion and written in standard English?

Reviewer #1: Yes

6. Review Comments to the Author

Reviewer #1: "On the one hand, for the cases of living in or visiting Hubei, we defined the incubation period as the interval between the lastest departure from Hubei and earliest symptom onset. Because these cases were not recorded the date of exposure to the

SARS-CoV-2." - I am sorry but this is wrong practice. The authors cannot do it, because it is not the incubation period. Incubation period is the time period from exposure to the virus to the illness onset. It does not equal to the time period from the latest departure date from Hubei to the symptoms onset. What the authors say is that some records of the data are censored. There are statistical methods to adress this and the authors need to do it properly.

Because a part of the results is incorrect, I cannot recommend this manuscript for acceptance and additional analysis is required.

7. PLOS authors have the option to publish the peer review history of their article (what does this mean?). If published, this will include your full peer review and any attached files.

Reviewer #1: No

---

## [Author Response · Author response to Decision Letter 1]

25 Sep 2020

We would like to thank the reviewers and editor for their help in our manuscript. We have realized that the available data do not support our estimation of the incubation period, although we have found an ideal model. We plan to estimate the incubation period in the next work, after getting more data. We are still happy to revise our manuscript according to the opinions of reviewers and editors.

---

## [Editor Report · Decision Letter 2]

16 Oct 2020

Pattern of COVID-19 in Sichuan Province, China: a descriptive epidemiological analysis

PONE-D-20-09556R2

Dear Dr. Feng,

We’re pleased to inform you that your manuscript, "Pattern of COVID-19 in Sichuan Province, China: a descriptive epidemiological analysis" (PONE-D-20-09556R2), has been judged scientifically suitable for publication and will be formally accepted for publication once it meets all outstanding technical requirements.

Kind regards,

Abdallah M. Samy, PhD

Academic Editor

PLOS ONE

---

## [Editor Report · Acceptance letter]

22 Oct 2020

PONE-D-20-09556R2 

Pattern of COVID-19 in Sichuan Province, China: a descriptive epidemiological analysis 

Dear Dr. Feng:

I'm pleased to inform you that your manuscript has been deemed suitable for publication in PLOS ONE. Congratulations! Your manuscript is now with our production department. 

Kind regards, 

on behalf of

Dr. Abdallah M. Samy 

Academic Editor

PLOS ONE